# Pre- and intra -COVID-19 trends of contraceptive use among women who had termination of pregnancy at Charlotte Maxeke Johannesburg Academic Hospital, Johannesburg South Africa (2010–2020)

**Kennedy Baffour-Duah**[1]*, **Gbenga Olorunfemi**[2], **Lusanda Shimange-Matsose**[1]

**1** Faculty of Health Science, Department of Obstetrics and Gynaecology, School of Clinical Medicine, University of the Witwatersrand, Johannesburg, South Africa, **2** Division of Epidemiology and Biostatistics, School of Public Health, University of the Witwatersrand, Johannesburg, South Africa

* baffduah@gmail.com

## Abstract

### Background

Contraception is a key prevention strategy for maternal morbidity and mortality and is an important component of postabortion care. The trend of contraceptive uptake can guide interventions among vulnerable individuals. The aim of the study was to determine the trends of immediate post-abortion contraceptive uptake among women who had induced abortion at a dedicated abortion centre in Johannesburg, South Africa.

### Method

This study was a retrospective cross-sectional and trend analysis of the contraceptive uptake among women who had legal termination of unwanted pregnancy at the Charlotte Maxeke Johannesburg Academic Hospital (CMJAH), from 1 January 2010 to 31 December 2020. We reviewed the nursing records and summaries of the induced abortion cases that were performed for the past eleven years from 1 January 2010 to 31 December 2020. The trends in the annual number of abortion seeking clients, annual contraceptive uptake rate (stratified by types), age of clients and gestational age at presentation were extracted into a spreadsheet for analysis. Join point regression modelling and Pearson's chi square were utilized to assess the trends and association between categorical variables. The trends before and during the Corona Virus disease(COVID-19) era were also compared.

### Results

In all, 12,006 clients had induced abortion over the study period. Nearly half (n = 5915, 49.26%) of the clients were aged 26–40 years, one tenth (n = 1157, 9.64%) being teenagers and one third (n = 4619, 38.47%) between age 19–25 years. There was a 16.3% annual increase in the number of abortion clients performed at the facility from 2010 to 2014 and a

**Data Availability Statement:** All relevant data are within the manuscript and its supporting information files.

**Funding:** GO is funded by GSK Africa Non-Communicable Disease Open Lab through the DELTAS Africa Sub-Saharan African Consortium for Advanced Biostatistics training programme. The views expressed in this publication are those of the author(s) and not necessarily those of GSK. GSK grant number D1702270-01.

**Competing interests:** the authors have declared that no competing interests exist.

gradual declining trend of about 9.2% per annum from 2014 to 2019. The overall postabortion contraceptive uptake rate was 74.5%. There was an initial 15.1% annual decline in contraceptive uptake per 100 clients from 2010 to 2014 and a subsequent increasing trend of about 11.1% per annum from 53.45 per 100 clients in 2014 to 98 per 100 clients in 2019. About 66.43% of the clients opted for injectable contraceptives. There was a reduction in the number of abortion seeking clients by 32.2% from 985 in 2019 to 668 in 2020 during the COVID-19 era. However, the contraceptive uptake was still high in 2020 (90.72%). There was no statistically significant relationship between the age group and the time periods. Thus, the distribution of the age group accessing the abortion services did not significantly change over the two time periods of 2010–2014 and 2015–2019. (P-value = 0.076).

## Conclusion

There was increased trends in postabortion contraceptive uptake among our clients from 2010 to 2020. Although there was reduced number of performed induced abortion during the COVID -19 era, the contraception uptake rate was still high during the COVID-19 era. About 6 out of every 10 clients accepted injectable contraceptives. More education is needed to improve uptake of other long-acting contraception that may not require frequent contact with the health facility.

## Introduction

The practice of contraception is an important prevention strategy for maternal mortality and morbidity [1]. Furthermore, modern contraceptive devices help to prevent morbidity and mortality that may occur from unsafe abortion by the prevention of unintended pregnancies [2].

Some countries such as South Africa have liberalized access to safe induced abortion. In contrast, countries such as Nigeria and Ghana have restrictive abortion laws [3]. It is expected that liberalization of access to induced abortion services can reduce unsafe abortion and its complications. Nonetheless, safe induced abortion can still have some degree of complications in countries that have liberalized abortion [4]. Hence, prevention of unintended pregnancies is a very important intervention for preventing the incidence of induced abortion and its sequelae. Indeed, counselling and access to family planning is also a core component of post abortion care [5,6]. Such contraceptive access can also prevent repeat episodes of unintended pregnancies. The WHO medical eligibility criteria support and encourage practitioners to offer appropriate contraceptive methods to patients after abortion [5]. Virtually all contraceptive methods can be utilized in the immediate post abortion period [7]. Moreover, it is the reproductive rights of all clients to be exposed to and counseled on all the various contraceptive methods [7]. Clients should then be allowed to freely make their choice in a non-judgmental approach.

Successive South African governments have initiated strategic measures aimed at promoting contraceptive use among the various key population [5]. Furthermore, the 1996 constitution of the republic of South Africa, Act 108 of 1996, promotes reproductive rights and the rights of access to reproductive health care [5]. South African government encourages the dissemination of information and counselling on contraception, sexuality and reproductive care in all hospitals in the country [5]. Furthermore, government supplies free contraceptives to all

clients at government owned hospitals. Despite these interventions, the contraceptive prevalence in South Africa according to the 2016 demographic and health survey is 60% [7]. Countries with similar interventions such as the United Kingdom, United States of America, Germany, and Australia have contraceptive prevalence of 84%, 74%, 69% and 68% respectively.

In January 2020, the WHO declared the outbreak of SARS-COV-2 as a public health emergency necessitating international concern [8]. South Africa declared its first case of Corona Virus disease (COVID-19) on 5[th] March 2020. As of June 10, 2020, there were 212,003 cases and 5,717 deaths recorded in Africa and South Africa had the highest burden of COVID-19, with 55,421 cases and 1210 deaths [9]. This pandemic can pose a threat to the sexual and reproductive health of women. However, the effect on contraception uptake and unintended pregnancy risk in women in South Africa was largely unknown. Indeed, Taylor et al projected that 15 million additional unintended pregnancies, leading to 28000 maternal deaths, can occur in one year if there is COVID-19-related disruptions in sexual and reproductive health services [10,11].

Post abortion contraceptive counselling and uptake is an important component of post abortion care, aimed at prevention of subsequent unintended pregnancies and induced abortion [5,6]. Virtually all contraceptive methods can be safely utilized by clients as post abortion contraceptives, but the trend of contraceptive uptake may be influenced by health system concerns such as distribution and availability of the various contraceptive types, and the individual experiences and preference [12]. Furthermore, an evaluation and documentation of the trends of post abortion contraceptive uptake will also help policy makers to evaluate the interventions aimed at reducing the unmet needs for contraception especially among post abortion clients. Additionally, the training of healthcare workers who offer these post abortion care services may be properly tailored to the needs of the clients based on the evidence from this study. We therefore aimed to determine the pre- and intra -COVID-19 trends of contraceptive use among women who had termination of pregnancy at Charlotte Maxeke Johannesburg Academic Hospital (2010–2020) in Johannesburg, South Africa.

## Method

### Study design

This study was a retrospective cross-sectional and trend analysis of the contraceptive uptake among women who had legal termination of unintended pregnancy under the South African laws at Charlotte Maxeke Johannesburg Academic Hospital (CMJAH), ward 195 from 1 January 2010 to 31 December 2020.

### Study site

The study was conducted at a dedicated induced abortion clinic (ward 195), department of Obstetrics and Gynecology of CMJAH Johannesburg. Ward 195 is a walk- in facility for clients who want to have termination of pregnancy in accordance with the South African laws on termination of pregnancy [13]. CMJAH is a quaternary Hospital located in Johannesburg. Johannesburg is the provincial capital of Gauteng with a population of 15.5 million in 2020. The Johannesburg city is mainly an urban city with a population of 5.6 million people as at 2020 [12].

The facility is manned by trained nurses and midwives who offer counselling and termination of pregnancy to the patients. There is always a supervising doctor on call who handles complicated cases. The facility has a counselling room, two functional procedure rooms and about 15 beds for the patients. There are 7 staffs (4 nurses, 2 doctors, 1 counsellor), and they

work from 8am to 5pm on Monday to Friday except on public holidays. All clients who report for termination of pregnancy are first seen and assessed by the doctor. They all get an ultrasound scan done to confirm and date the pregnancy. The patients are then referred to see the counselors, who together with the medical team try to understand the reasons for the termination and offer options if any for the clients. They are also counselled for contraception and offered their preferred option following the procedure. Where the patients do not get their preferred contraceptive, they are either referred to their local clinic or asked to report another day for it. But such patients are advised to use a barrier method until they are seen at their local clinics for the preferred choice. Manual vacuum aspiration with Karman syringe is utilized for most cases. However, some cases may require evacuation of uterus in theatre when there is incomplete miscarriage. Some of the cases of incomplete miscarriage are also managed with MVA or misoprostol. Following the procedure, the patients are sent to the recovery ward where they stay for 10–15 minutes for monitoring and observation and then given their preferred contraceptive. The facility usually performs an average of 100 induced abortions every month.

## Study procedure

We reviewed the nursing records and summaries of the induced abortion cases that were performed for the past eleven years from 1 January 2010 to 31 December 2020. The annual number of patients who presented for termination of pregnancy, annual number of patients who had contraception afterwards and the various methods that were accepted were extracted into a spreadsheet for analysis. Furthermore, the annual age group of the patients, the gestational age at presentation were also recorded

## Ethical considerations

Ethical approval was obtained from the Research and Ethics Committee (Medical) of the University of Witwatersrand before the commencement of the study. (Ethics approval number: (M200802). Permission was also obtained from the Chief Executive Officer of CMJAH before the commencement of the study. The study was a retrospective cross-sectional study. We reviewed the nursing records and summaries of the induced abortion cases that were performed for the past eleven years from 1 January 2010 to 31 December 2020. The annual number of patients who presented for termination of pregnancy, annual number of patients who had contraception afterwards and the various methods that were accepted were extracted into a spreadsheet for analysis. Confidentiality of the data was ensured as names and other identifiers were not utilized. The data obtained was aggregated and fully anonymized. The ethics committee waved the requirements for informed consent from the clients to have the data from the nursing records used in the research.

## Statistical analysis

For the trends studies, the annual contraceptive uptake rate from 2010 to 2020 was calculated by dividing the number of clients who had contraceptives by the total number of post abortion clients. The annual contraceptive incidence was stratified by contraceptive types. The annual contraceptive uptake rate was reported per hundred or per thousand abortion cases for ease of comparison after respectively multiplying the rate by 100 or 1000.

Proportion of age group, gestational age at presentation ($< 13$ weeks and $\geq 13$ weeks), contraceptive uptake and contraceptive types was compared between the periods 2010–2014 and 2015–2019 using the Pearson's Chi square test. The above-mentioned variables were also

compared between 2019 (pre-COVID-19 era) and 2020 (Intra COVID-19 era) using Pearson's Chi-square.

The annual trends in contraceptive uptake were plotted on graphs to show the pattern (whether increasing or decreasing). Join point regression modelling (Version 4.7.1, National Cancer Institute https://surveillance.cancer.gov/joinpoint/) was used to determine the trends in abortion and contraceptive rate. The join point regression was used to run Log-linear model with one maximum join point and 4499 permutation tests to obtain the trends. The Average annual percent change (AAPC) was calculated. A statistically significant positive or negative AAPC was assumed as a statistically significant increase or decrease trends respectively. AAPC between -0.5 to +0.5 with no statistical significance is assumed as a stable trend.

Based on the AAPC obtained from Join point regression from 2010 to 2019, the number of abortions for year 2020 (intra-COVID-19 era) was predicted using the formula

AAPC = Number of abortions for 2019 –predicted number of abortions for 2020/ Number of abortions for 2019. (1)

Making predicted number of abortions for 2020 as the subject of the formula:

Predicted number of abortions for 2020 = (Number of abortions for 2019) * (1 –AAPC)

Furthermore, a linear regression was conducted with the annual number of abortions as the outcome and the year as the explanatory variable. An equation of the line was obtained. The equation of the line was utilized to predict the 2020 abortion numbers. The mean absolute percent error (MAPE) between the predicted and the actual abortion numbers before 2020 was calculated. Similar analysis was conducted for contraceptive uptake. Statistically significant P-value was assumed to be P-value < 0.05. Statistical analysis was done using Stata Version 17 (StataCorp, USA) software.

## Results

Over the 11 -year study period (2010–2020), 12,006 clients had induced abortion at the centre. However, 11,338 clients had induced abortion at the centre from 2010–2019 (pre-COVID-19 era) and the average annual abortions performed between 2010 and 2019 was 1,133 procedures per annum. Nearly half of the clients were aged 26–40 years (n = 5589, 49.04%) and about 47.71% of the clients were younger than 26 years. More than half of the clients that had abortion procedures at the centre had second trimester pregnancy (n = 6485, 56.86%).

From Table 1, the number of abortion procedures were nearly the same between the two time periods of 2010–2014 (n = 5641 (49.75%) and 2015–2019 (n = 5697 (50.25%).

There was no statistically significant relationship between the age group and the time periods. Thus, the distribution of the age group accessing the abortion services did not significantly change over the two time periods of 2010–2014 and 2015–2019. (P-value = 0.076).

**Table 1. Comparison of number of abortions, age and gestation at termination across two time periods.**

| Variable | Total (2010–2019) N = (%) | 2010–2014 N = (%) | 2015–2019 N = (%) | P-value |
|---|---|---|---|---|
| Number of abortion cases | 11338 | 5641 (49.75) | 5697 (50.25) | |
| Age (Years) | | | | |
| 12–18 | 1,082 (9.49) | 577 (10.23) | 505(8.86) | 0.076 |
| 19–25 | 4,356 (38.22) | 2156 (38.22) | 2200 (38.62) | |
| 26–40 | 5,589 (49.04) | 2770 (49.10) | 2819 (49.48) | |
| ≥40 | 369 (3.24) | 195 (3.46) | 174 (3.05) | |
| Gestational age at abortion (weeks) | | | | |
| <13 | 4,921 (43.14) | 1218 (21.40) | 3703 (64.79) | < 0.001 |
| ≥13 | 6,485 (56.86) | 4473 (78.60) | 2012 (35.21) | |

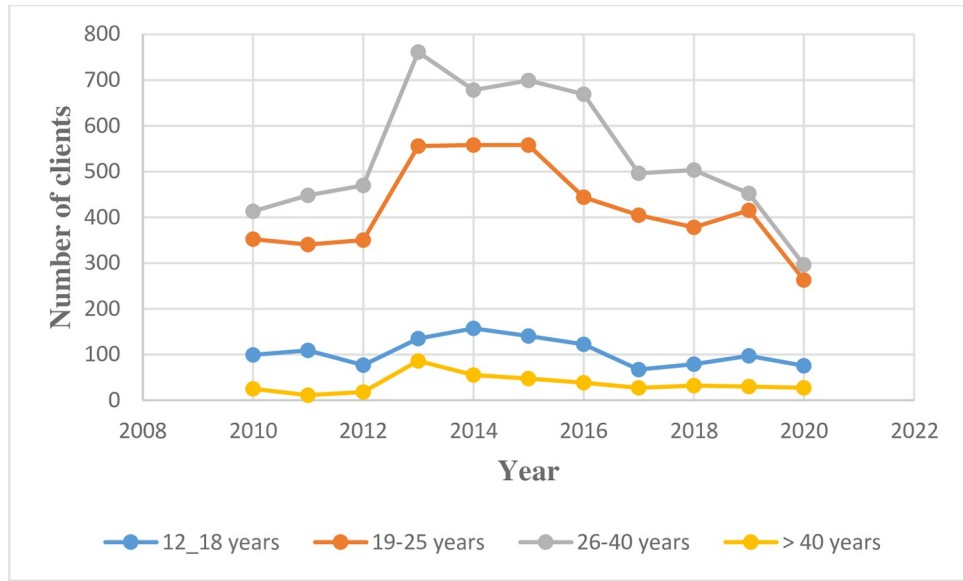

**Fig 1. Trends in the annual number of clients for Abortion procedures stratified by age.**

### Trends in age of clients

Furthermore, all the age groups had similar annual trends of nearly stable trends from 2010 to 2013. There was a subsequent abruptly increased trends among women in the 26–40 years and 19–25 years category from 2012 to 2013. Afterwards, a gentle downward trend occurred till 2019. However, there was a sharp decrease between 2019 and 2020 among all the age groups. (S1 Table and Fig 1).

### Trends in gestational age at abortion

During the first 5-year period (2010–2014), more than three-quarters (78.6%, n = 4473) of the abortion performed occurred among women with second trimester pregnancy. However, during the later time-period (2015–2019), the prevalence of second trimester abortion dropped by more than 50% to 35.2% (n = 2012). Indeed, there was no client with second trimester pregnancy from 2017 to 2020. (See S1 Table and Fig 2).

### Pre and intra-covid number of abortions, age and gestational age of clients and contraceptive uptake

From Table 2, the number of abortion cases reduced by 32.2% from 985 clients in 2019 (pre-COVID-19 era) to 668 clients in 2020 (Intra-COVID-19 era). There was no statistically significant difference in the age category that presented for induced abortion care during the pre- and intra COVID-19 era. (P-value 0.252). All cases of induced abortion were less than 13 weeks gestation in 2019 and 2020.

From Table 3, The overall contraceptive prevalence rate from 2010–2019 was 73.51% (n = 8335/ 11338). Of the 8,335 clients that accepted post abortion contraceptive, about nine-tenth (89.13%, n = 7429/8335) accepted injectable contraceptives while about 9.6% (n = 804/ 8335) accepted oral contraceptives. Very few clients accepted implants (0.67%, n = 56/8335) and intrauterine contraceptive device (0.55%, n = 46/8335).

The number of patients who had contraception following the abortion procedures were nearly the same between the two time periods of 2010–2014 and 2015–2019. Thus, the

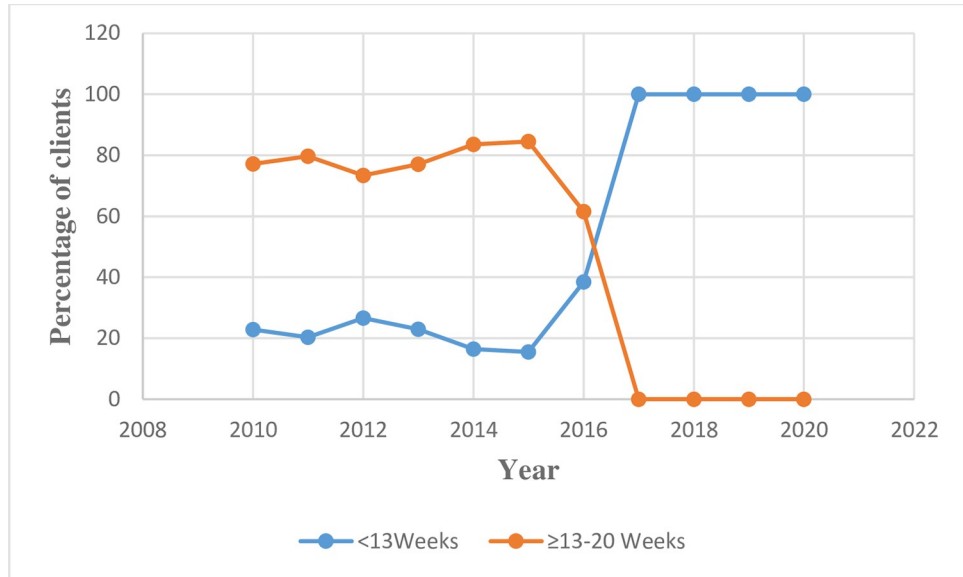

**Fig 2. Trends in the annual proportion of clients for Abortion procedures stratified by proportion gestational age at presentation.**

prevalence of immediate post abortion contraceptive uptake among the clients within the two time periods remained the same (about 73%.). The dominant contraceptive commodity that was accepted by the clients was injectable contraceptive (66.43%, N = 7975). While a few had oral contraceptive pills (7.01%, n = 842), implants (0.54%, n = 65) and intrauterine contraceptive device (0.44%, n = 53). Comparing the two-time period, about two-third of the clients accepted injectable contraceptives during the 2010–2014 (64.92%) and 2015–2019 (66.12%) time periods.

Whereas all the contraceptive methods recorded some gains in the second period from 2015–2019, the use of oral contraceptives however declined by almost 50% from 10.55% (n = 595) in 2010–2014 period to 5.42% (n = 5.42%) in the 2015–2019 period. None of the clients accepted the other long-acting reversible contraceptives such as the intrauterine contraceptive in the first period from 2010–2014. However, in the second period from 2015–2019 their use started increasing although not yet significant as both recorded less than 1%.

Table 4 summarizes the abortion cases performed and the contraceptive uptake in 2019 (pre-COVID-19) and 2020(COVID-19 era). Notwithstanding that there was a decline of about

**Table 2. Comparison of number of abortions, age and gestation age at termination across the pre-COVID-19 and intra-COVID-19 era.**

| Variable | Total (2010–2020) N = (%) | 2019 N = (%) | 2020 N = (%) | P-value |
|---|---|---|---|---|
| Number of abortion cases | 12006 | 985(59.59) | 668(40.41) | |
| Age (Years) | | | | |
| 12–18 | 1157(9.64) | 97 (9.76) | 75 (11.83) | 0.252 |
| 19–25 | 4619(38.47) | 415 (41.75) | 263 (41.48) | |
| 26–40 | 5915(49.26) | 452(45.47) | 269(42.43) | |
| ≥40 | 386(3.22) | 30 (3.02) | 27 (4.26) | |
| Gestational age at abortion (weeks) | | | | |
| <13 | 5589(46.55) | 985(100.00) | 668(100.00) | - |
| ≥13 | 6475(53.45) | 0(0.00) | 0(0.00) | |

**Table 3. Comparison of 5-year periods of pattern of contraceptive uptake.**

|  | Total (2010–2019) N = (%) | 2010–2014 N = (%) | 2015–2019 N = (%) | P-value |
|---|---|---|---|---|
| Number of abortion | 11338 | 5641(49.75) | 5697(50.25) | - |
| Total contraceptive uptake |  |  |  |  |
| Yes | 8335((73.51) | 4157(73.69) | 4178(73.34) | 0.668 |
| No | 3003(26.49) | 1484(26.31) | 1519(26.66) |  |
| Types of contraceptive |  |  |  |  |
| Oral Contraceptive pills | 804(7.09) | 595(10.55) | 309(5.42) | < 0.001 |
| Injectable contraceptive | 7429(65.52) | 3662(64.92) | 3767(66.12) |  |
| Implants | 56(0.49) | 0(0.00) | 56(0.98) |  |
| Intrauterine device | 46(0.41) | 0(0.00) | 46(0.81) |  |
| None | 3003(26.49) | 1484 (26.31) | 1519(26.66) |  |

32.2% in the abortion cases performed, the pre and intra COVID-19 eras still recorded similarly high contraception prevalence rate of 91.78% and 90.72% respectively. (p = 0.453). Though not reaching statistically significant difference, all the reversible long-acting contraceptives recorded some increase in uptake rate from 2019 to 2020. injectable contraceptive uptake was accepted by about four-fifth of the clients in both 2019 (80.71%) and 2020 (81.74%) respectively. Likewise, there was a minimal increment in the other long-acting reversible contraceptive methods such as the Implanon (1.12% to 1.35%) and intrauterine contraceptive devices (0.61% to 1.05%). However, uptake of oral contraceptive pill declined from 8.12% in 2019 (pre-COVID-19) to 5.69% in 2020 (intra- COVID-19).

The overall trend in the contraceptive mix is presented in Fig 3 and S2 Table.

Fig 3, (S1 Fig) below shows a gradual decline in uptake of injectable contraceptives from 2011 to 2012. Thereafter the uptake started increasing and remained high until 2015 when there was another dip in its uptake. The uptake of oral contraceptives on the other hand saw a gradual increase from 2010 reaching its peak in 2012. From 2013, the uptake of oral contraceptives persistently declined reaching its lowest point in 2016. The uptake of the oral contraceptives started increasing again from 2017 until 2020 when it experienced another decline.

The uptake of other reversible long-acting contraceptives like Mirena and Implanon in the facility started after 2016 and has since not experienced any significant trend in their uptake.

**Table 4. Comparison of pattern of Contraceptive uptake during the pre-COVID-19 and intra-COVID-19 era.**

|  | Total (2010–2019) N = (%) | 2019 N = (%) | 2020 N = (%) | P-value |
|---|---|---|---|---|
| Number of abortion | 11338 | 985(59.59) | 668(40.41) |  |
| Total contraceptive uptake |  |  |  |  |
| Yes | 8335(73.51) | 904(91.78) | 606(90.72) | 0.453 |
| No | 3003(26.49) | 81(8.22) | 62(9.28) |  |
| Types of contraceptive |  |  |  |  |
| Oral Contraceptive pills | 804(7.09) | 80(8.12) | 38(5.69) | 0.286 |
| Injectable contraceptive pills | 7429(65.52) | 795(80.71) | 546(81.74) |  |
| Implants | 56(0.49) | 11(1.12) | 9(1.35) |  |
| Intrauterine device | 46(0.41) | 6(0.61) | 7(1.05) |  |
| None | 3003(26.49) | 81(8.22) | 62(9.28) |  |

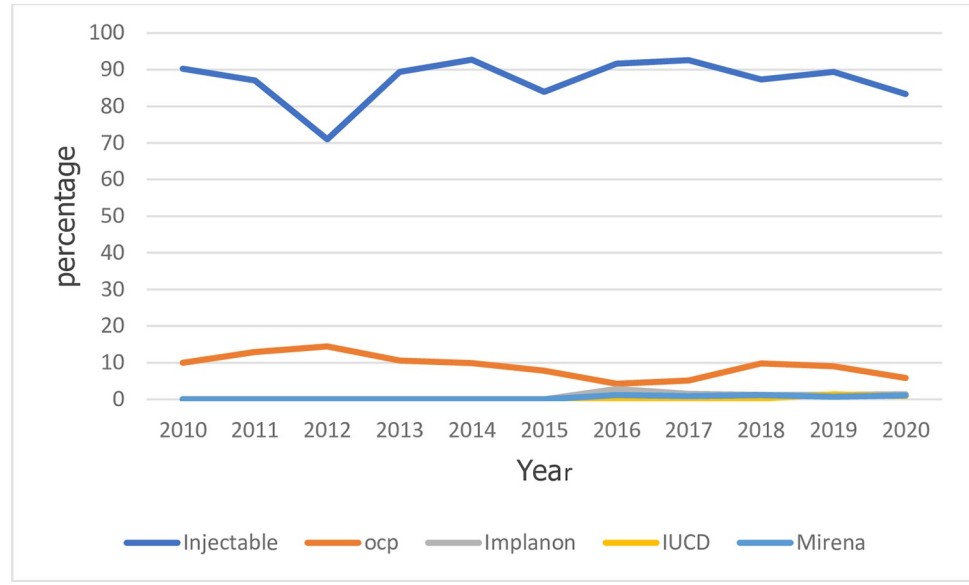

**Fig 3. Graphs showing the annual proportion of the various contraceptive uptake within the facility.**

## Trends in annual number of abortion procedures and contraceptive uptake prevalence

From Fig 4 and Table 5, the number of abortion procedures performed increased gradually from 877 procedures in 2010 to 929 procedures in 2012. Subsequently, there was an abrupt

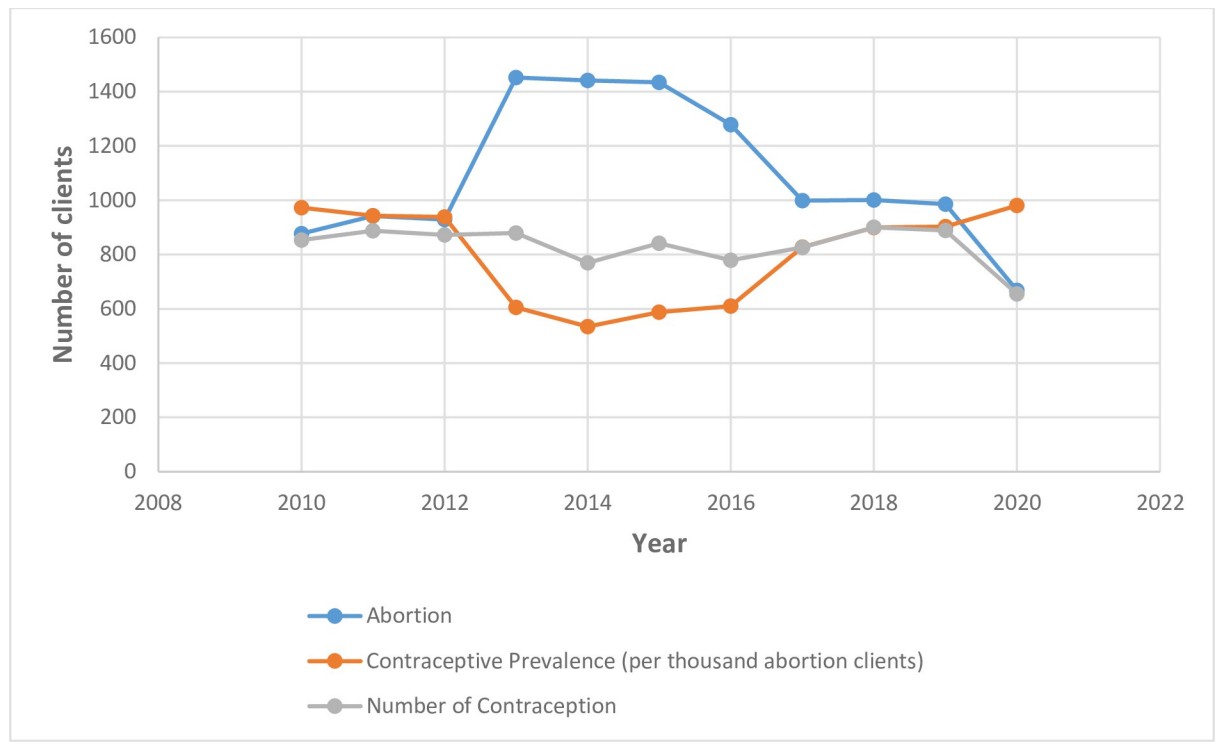

**Fig 4. Trends in the number of clients who had induced abortion, number of clients who had contraceptive uptake and the prevalence of contraceptive uptake per thousand abortion cases.**

**Table 5. Annual trends in overall contraceptive uptake.**

| Years | Abortion | | Contraception | | | |
|---|---|---|---|---|---|---|
| | Number of abortion clients | £APC (%) | Number of clients who had post-abortion contraceptive | Prevalence of contraceptive uptake. Per hundred abortion cases (%) | £Prevalence of contraceptive uptake (per thousand abortion cases) | £APC |
| 2010 | 877 | - | 853 | 97.26 | 972.6 | - |
| 2011 | 942 | +7.41 | 888 | 94.27 | 942.7 | -3.07 |
| 2012 | 929 | -1.38 | 872 | 93.86 | 938.6 | -0.43 |
| 2013 | 1452 | +56.30 | 879 | 60.54 | 605.4 | -35.50 |
| 2014 | 1441 | -0.76 | 770 | 53.45 | 534.5 | -11.71 |
| 2015 | 1434 | -0.49 | 842 | 58.72 | 587.2 | +9.86 |
| 2016 | 1278 | -10.88 | 779 | 60.95 | 609.5 | +3.80 |
| 2017 | 999 | -21.83 | 826 | 82.68 | 826.8 | +35.65 |
| 2018 | 1001 | +0.20 | 900 | 89.91 | 899.1 | +8.74 |
| 2019 | 985 | -1.60 | 889 | 90.25 | 902.5 | +0.38 |
| 2020 | 668 | -32.18 | 655 | 98.05 | 980.5 | +8.64 |

£Annual percent change.

increase in the number of abortion procedures to 1452 cases in 2013 and plateau to 2015 (1434 cases). Afterwards, there was a decline from 2015 to 999 procedures in 2017. Another plateau occurred from 2017 to 2019 (985 cases). However, contraceptive methods were provided for between 770 and 900 clients from 2010–2019 with no obvious graphical relationship with the trends in Abortion cases.

There was a slight decline in contraceptive uptake from 97.2% in 2010 to 93.86% in 2012. There was a subsequent deep in contraceptive uptake from 2012 to 58.72% in 2015. Afterwards, the contraceptive uptake consistently increased from 2015 to 98.05% in 2020

From Fig 5, join point regression modelling revealed two trends in the annual number of abortion cases that were managed at the centre. The first trend was an increase in annual number of abortions from 877 in 2010 to 1441 abortion cases in 2014 at a rate of 16.3% per annum (AAPC = 16.3%, 95% CI: 1.8% -32.8%, P-value < 0.001). The second trend in annual number of abortions was a statistically significant decline from 1441 abortion cases in 2014 to 985 abortion cases in 2019 at annual rate of decline of 9.2% per annum (AAPC = 9.2%, 95% CI: -17.1% to -0.6%, P-value < 0.001) Table 6.

There were non-statistically significant initial trends in annual number of contraceptive commodities from 2010 to 2016 (P-value = 0.2) and a subsequent non-statistically significant increased trend in annual number of methods from 2016 to 2019 (P-value = 0.3) (Fig 6). With respect to trends in contraceptive prevalence per hundred abortion clients, there was an initial statistically significant decreased trends in annual contraceptive uptake per hundred abortion clients at an average of 15.1% per annum from 97.3% in 2010 to 53.5% in 2014 (AAPC = 15.1%, 95%CI: -26.4 to -2.1, P-value < 0.001). Subsequently, there was a statistically significant increase in the contraceptive uptake per hundred abortion clients at 11.1% per annum from 53.5% in 2014 to 98.1% in 2019 (AAPC = 11.1%, 95%CI: 0.0 to 23.4%, P-value < 0.001) Fig 6 and Table 6.

## Predicted deviation of trends in abortion procedures and contraceptive uptake rate during the COVID-19 era (2020)

From Table 7, it is predicted that based on the join point trends with AAPC of 9.2% per annum from 2014 to 2019, 894 abortion procedures (95% CI: 817–979) should have been

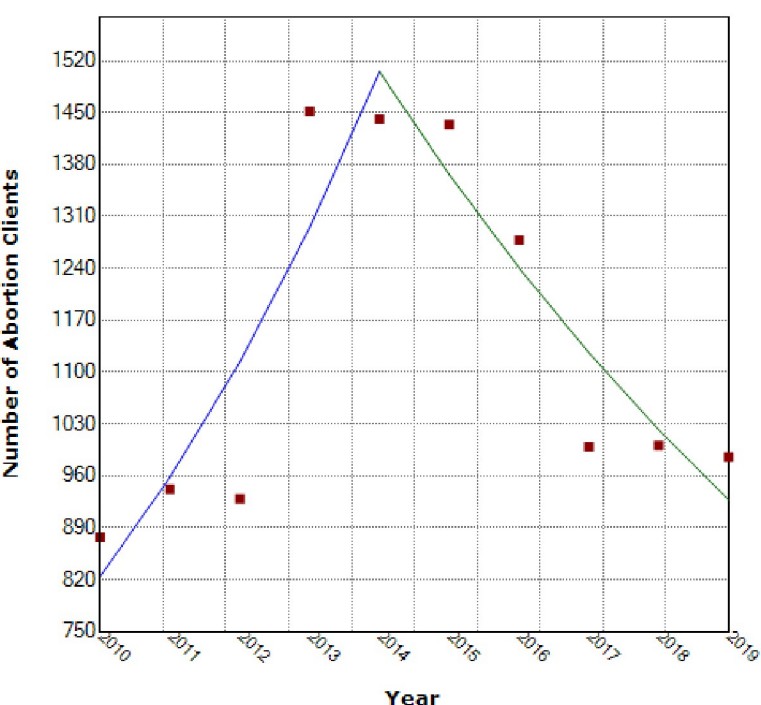

**Fig 5. Join point trends of the annual number of abortion procedures (2010–2019).**

performed in 2020. Additionally, 804 procedures (with a MAPE of 5.5%) was predicted to be performed in 2020 from linear regression modelling. The equation of the prediction based on linear regression is:

Abortion = 223465.6–110.2 * Year.

The coefficient of determination ($R^2$ = 0.87) of the prediction line was high (P-value = 0.0067) and this showed that the model can explain about 87% of the variation in abortion trends.

However, the actual abortion cases for 2020 (intra-COVID-19 era) was 668 procedures, which was lower than the predicted value of 894 by about 33.8%

**Table 6. Join point regression estimates of the trends in overall abortion and contraceptive uptake (2010–2019).**

| Join point estimates | Abortion procedures | | Number of Contraceptive commodities | | Prevalence of contraceptive uptake | |
|---|---|---|---|---|---|---|
| | Trend I 2010–2014 | Trend II 2014–2019 | Trend I 2010–2016 | Trend II 2016–2019 | Trend I 2010–2014 | Trend II 2014–2019 |
| AAPC | 16.3 | -9.2 | -1.7 | 4.3 | - 15.1 | 11.1 |
| 95%CI | 1.8–32.8 | -17.1 to -0.6 | -4.5 to 1.3 | -4.3 to 13.7 | -26.4 to -2.1 | 0.0 to 23.4 |
| P-value | <0.001* | < 0.001* | 0.2 | 0.3 | <0.001* | < 0.001* |
| Comment | Statistically significant increasing trends | Statistically significant decreasing trends | Non-statistically decreasing trends | Non-statistically significant increasing trends | statistically significant decreasing trends | statistically significant increasing trends |

*Statistically significant at P-value< 0.05; CI: Confidence interval; Average annual percent change.

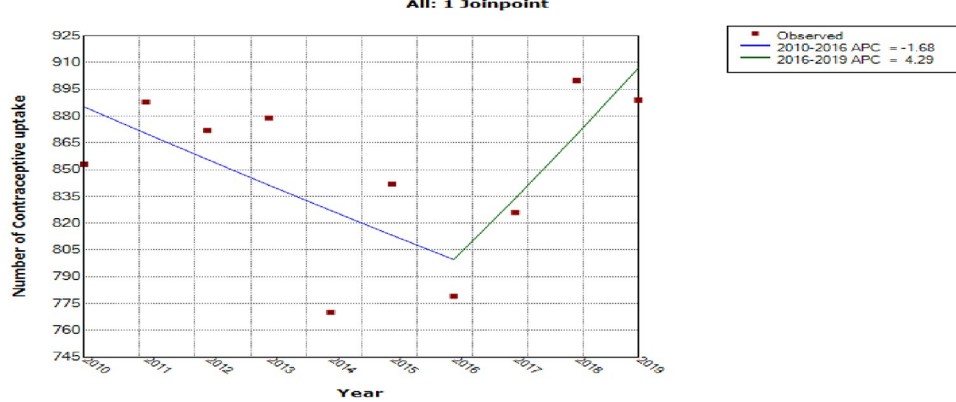

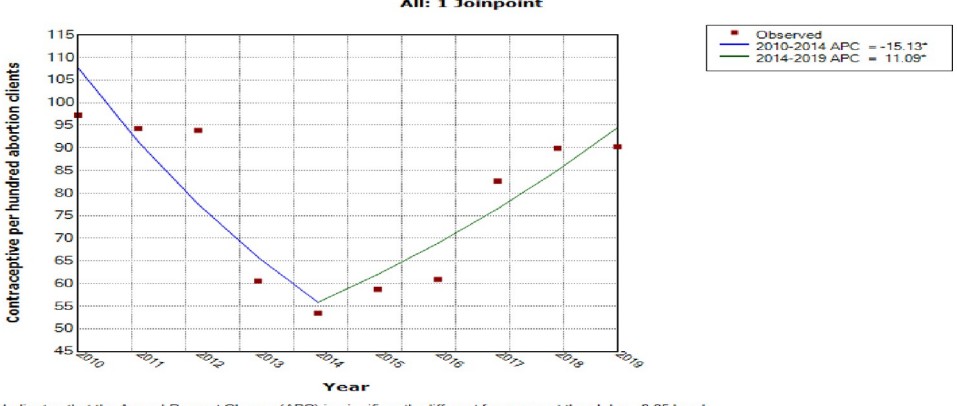

**Fig 6. Join point trends of the annual number of contraceptive uptake (2010–2019).**

Table 7 also showed that the predicted value of contraceptive uptake was 100.27% and 102.6% based on join point regression modelling and the linear regression modelling. The equation for the prediction line is:

Contraceptive prevalence = 8.55 * Year -17177.12 with a high coefficient of determination. ($R^2$ = 0.9091, P-value = 0.0032). This suggest that the predicted contraceptive uptake should be at the maximal level (100%) by 2020 (Intra-COVID-19 era). The actual contraceptive prevalence was just minimally lower than the predicted value (100.27% Vs 98.05%) (S3 Table).

## Discussion

This study was conducted to determine the trends and pattern of contraceptive uptake among women who had termination of pregnancy in one of the quaternary academic hospitals in

**Table 7. Predicted and actual cases of abortion and contraceptive uptake during the COVID-19 era (2020).**

| Prediction parameters | Abortion procedures | Contraceptive uptake | Prevalence of contraceptive uptake (%) |
|---|---|---|---|
| §Predicted model I | 894 | 927 | 100.27 |
| 95%CI of predicted model I | 817 to 979 | 850 to | 90.25 to 111.37 |
| ^Predicted model II | 804 | 916 | 102.6 |
| §MAPE | 5.48% | | 5.69% |
| α P-value | 0.93 0.0067 | 0.81 0.05 | 0.95 0.0032 |
| Actual /observed | 668 | 655 | 98.05 |
| Comment | Less abortion clients by 136 to 226 than predicted (-33.83%) | | About 2.2% to 3.9% less than predicted |

§MAPE: Mean absolute percent error

α: Pearson's correlation coefficient

^Predicted model II: Predicted value based on linear regression trends from 2014–2019

§ Predicted model I: Predicted value based on join point regression model.

Johannesburg from 2010 to 2020. To our knowledge, this is the first study to evaluate the trends of contraceptive uptake among abortion seekers in South Africa.

## Summary of findings

We found that more than a thousand clients per annum utilized the induced abortion services of the Hospital, and more than half of the clients presented in the second trimester. However, in the last four years of the study (2017–2020) there was no second trimester abortion performed in the facility. Nearly half of the clients were aged 26–40 years while about one-third of the clients were within the age range of 20–25 years. About one-tenth of the abortion seekers were teenagers younger than 19 years. There was a 16.3% annual increase in the number of abortion clients performed at the facility from 2010 to 2014 and a gradual declining trend of about 9.2% per annum from 2014 to 2019. The overall prevalence of contraceptive uptake was 74.51% during the study period with nine out of ten clients accepting injectable contraceptives. Nearly one-tenth of those on contraceptives accepted oral contraceptives pills. Other contraceptive types such as Intrauterine contraceptive device and implants are getting accepted in the last 5 years of the study (2015–2020). There was an initial 15.1% annual decline in contraceptive uptake per 100 clients from 2010 to 2014 and a subsequent increasing trend of about 11.1% per annum from 53.45 per hundred clients in 2014 to 98 per 100 hundred clients in 2019. We also found that the predicted number of abortion procedures during the COVID-19 era was about 20% (230 cases lesser) lesser than the actual number of abortion cases that was managed while there was just a slight difference of about 2% between the predicted contraceptive uptake rate (100.3%) and the actual contraceptive uptake rate (98.1%) during the COVID-19 era.

A finding that more than one thousand women sought and had safe induced abortion at our single centre study highlights the importance of providing such services to reduce morbidity and mortality related to unsafe abortion. Unfortunately data on the number of abortions performed in the other public facilities in the province are not readily available for comparison [13]. Indeed, our study revealed that thousands of women with unintended pregnancies would have gone ahead to procure unsafe abortion with its attendant complications, if safe induced abortion were not legalized in South Africa and the facility for safe abortion was not readily available. The gradual decline in number of clients seeking abortion by about 9% per annum from 2014 to 2019 may suggest that public health interventions to increase contraceptive

prevalence in the country might have led to a reduction in unintended pregnancies [14,15]. On the other hand, the decline may be related to health system or institutional concerns. A decentralization of the abortion services in the country as stipulated in the health act 2003 and its subsequent implementation in the various provinces [16] might have led to the reduction in the number of clients visiting our centre. Furthermore, patient's satisfaction in the service rendered may also impact on the number accessing the service [17]. Further studies are necessary to unravel these reduced abortion trends.

Nearly half of the clients at the facility were aged 26–40 years. Closely followed by the women aged 19–25 years (38.22%). This finding supports studies that women aged 20–29 years have the highest proportion of abortions in most countries [18]. It is worthy of note that teenagers younger than 19 years constituted about 10% of the abortion seekers among our cohort. During the teenage age, girls may not be fully aware of the reproductive systems and may wish to experiment. Furthermore, the teenagers may also be victim of rape and incest. Approximately 1 in every 10 teenagers get sexually abused before age 18 [19]. Although, abstinence should be encouraged among teenagers, our finding underscores the need for sex education and provision of easy access to modern contraceptives among sexually active teenagers to prevent unintended pregnancies [20]. If the facility for safe abortion was not available to this large number of teenagers, they may either resort to procuring unsafe abortion or drop out of school to take care of the baby. Thus, this facility and others offering safe abortion services might have contributed largely to reducing school drop-out rate among girl child in South Africa that would have been occasioned by unwanted pregnancies.

We found that the post abortion contraceptive uptake rate among the clients at our facility was about 74.5% which is higher than the national average of about 60% for sexually active women according to the 2016 South African Demographic and health survey report [7]. Indeed, the contraceptive uptake was increased to about 98% in 2020. Similarly, Benson et al found that on the post abortion contraceptive uptake among young women who had induced abortion across 10 countries in sub-Saharan Africa and Asia was about 77% [14]. However, the contraceptive uptake rate was higher than the reported post-abortion contraceptive uptake 61% form a study in Ethiopia. The high contraceptive uptake rate among the cohort of abortion seekers may be because of the impact of post-abortion counselling offered the women and the strong urge to prevent another unintended pregnancy [21]. The high contraceptive uptake among our cohort of clients may also be related to social desirability as the clients may believe the medical personnel will not be happy and ready to assist if she did not accept a contraceptive method after counselling. The process of counselling must be sensitive to protect the autonomy of the patients. Post abortion family planning counselling should inform the clients that they have the right to choose freely if they want to have post-abortion contraceptive without cohesion (no matter how subtle).

Our study showed that the contraceptive prevalence rate among women who had abortion in the facility increased at a rate of 11.1% per annum from 2014 to 2020 and the contraceptive uptake rate was about 98 per 100 abortion clients in 2020. Our study although agrees with their finding of decline in the utilization of abortion services by Adelekan et al, but we noted increase in the uptake of reversible long acting contraceptive use(copper IUCD, Mirena, Injectables and Implanon) and decline in the uptake of oral contraceptives which are in direct contrast to their study in 2020 [15].

Injectable contraceptive remained the most sought-after contraceptive method among the clients at our centre as about nine out of every ten clients accepted this method. On the other hand, nearly one-tenth of clients that accepted contraceptives opted for combined oral contraceptives pills, the use of oral contraceptives declined over the study period while the injectable contraceptive uptake was stable. Our result was in contrast to the report of Benson et al and

Adelekan et al that reported oral contraceptive pills as the most commonly preferred contraceptive method followed by condoms and injectables [14,15]. Other contraceptive types such as Intrauterine contraceptive device (IUCD) and implants are getting accepted in the last 5 years of the study (2015–2020). Although, the uptake of other long-acting reversible contraceptives (aside from injectables) seems to increase lately, their uptake rate of < 2% among the cohort in the last five years (2016–2020) is still very low. It appears that injectable contraceptives were more readily available in South Africa as compared to other long acting reversible contraceptives [22]. Furthermore, the use of less effective contraceptive methods, such as the oral contraceptive pill (OCP) and the persistent underutilization of long-acting contraceptive methods (IUCD, hormonal implants) may be an indication of limited access to them, or limited access to appropriate information on all the spectrum of contraceptive options. If the patronage of long-acting reversible contraceptives improves, it will help to reduce the contraceptive failure rates and hence reduce unwanted pregnancies. Thus, efforts should be geared towards ensuring adequate contraceptive mix among abortion seekers to protect their reproductive health rights.

The COVID-19 pandemic seems to have had some impact on the total number of abortion procedures that ought to have been performed in the facility in 2020 based on predictions modelling of the previous trends. Thus, abortion procedures in the facility dropped by 32% in 2020 and about 226 lesser abortion procedures were performed based on the predictions from the previous 10-year trend from 2010 to 2019. In March 2020 there was a national lockdown and subsequent redistribution of staff across Gauteng hospitals, allocation of more resources to covid-19 related issues, fear and anxiety among the population including health workers might have affected the patronage for induced abortion at our facility [15]. Nonetheless, the contraceptive prevalence rate was still very high although it slightly decreased from 91.78% in 2019 (pre-COVID-19 era) to 90.71% in 2020 (COVID-19 era). However, our suspicion is that women sought abortion at other facilities that were easily accessible to them during the lockdown and other restrictive period of the COVID -19 era.

Decline in reproductive health consultations such as family planning visits and contraceptive use were reported in Liberia, Sierra Leone and Guinea during the Ebola outbreaks and six months after the epidemic, indicating that epidemics can have deleterious effects on reproductive health [15]. Fear of contracting the disease, mandatory curfews, closure of borders and transportation routes, are some of the reasons that prevented people from obtaining medical services including reproductive health services [9]. Efforts to ensure that women can still access induced abortion service during COVID-19 era is very important to reduce the epidemic of morbidity and mortality from unsafe abortion [11]. Innovative use of technology such as online booking and calling to register for the procedure to avoid overcrowding during an epidemic can be considered. Further research is needed to review how to ensure reproductive services such as induced abortion services are provided during the pandemic. Reproductive health services are a critical aspect of human rights. Therefore, medical and social support and access to these services should be equitably available during epidemics [9].

About 90.71% of the clients opted for injectable contraceptives during the COVID-19 era, however, some women (<2%) are now opting for longer acting contraceptive methods such as the copper intrauterine contraceptive device (Cu IUCD), Levonorgestrel-releasing intrauterine system (Mirena) or Etonogestrel implant (Implanon) during the COVID-19 era. Such a trend to utilize IUCD and Implanon may be very imperative during the COVID-19 era to reduce the frequent visits and contact to the health facilities that occur with injectable contraceptives. However, we found some decline in the uptake of oral contraceptive (OCP) from 8.12% in 2019 to 5.69% in 2020. Our result is contrary to the report by Adelekan et al conducted in

March-April 2020 that showed an increased prescription of OCP during the lockdown in Gauteng province of South Africa [15].

We observed that the prevalence of second trimester abortion dropped in the facility by more than 50% from 4473 in 2010–2014 period to 2012 during the 2015–2019 period. Indeed, there was no client with second trimester abortion who had induced abortion from 2017 to 2020. This finding may suggest that the incidence of clients with unintended or unwanted second trimester pregnancies are getting rare in South Africa because many facilities are readily available in the country to offer safe termination of the pregnancy as early as possible [13]. Moreover, clients may be more aware of the increased complications of second trimester induced abortion and tends to avoid same. However, a study conducted in referral hospital in Ethiopia, reported that a third of abortion performed in 2020 was for women in the second trimester [23]. Nevertheless, further research to know how or where clients who opted for second trimester induced abortion are being managed is necessary. Otherwise, clients in desperate need of termination of second trimester pregnancy might resort to clandestine unsafe abortion which could cause increased maternal morbidity and mortality. Indeed, the target maternal mortality ratio of less than 100 per 100,000 for Gauteng province [15] may be adversely affected if clients do not have access to legal and safe second trimester induced abortion services.

## Strength and limitations of the study

A limitation of the study was that the data available at the facility did not capture most of the client's demographic characteristics. However, this study is the first to document the trends of contraceptive uptake among women who had abortion at our centre.

## Conclusion

Our study showed that although there was decline in the annual number of induced abortion performed at our centre, the post abortion contraceptive uptake is very high. About 66.43% of the client accepted injectable contraceptives. More education is needed to improve uptake of other long-acting contraception. Long-acting reversible contraceptive should be encouraged to reduce the number of contacts with health facilities on account of family planning consultations during the COVID-19 era. Second trimester induced abortion was not performed in the last four years of the study. Access to second trimester induced abortion services should be available for those that may desperately need such service.

## Recommendations

More education is needed to improve uptake of long-acting contraception. Hospital supply chain of other contraceptive options should be strengthened. The demographic characteristics of the clients who procure abortion services should be adequately captured. Adequate contraceptive mix should be made available for post-abortion family planning counselling and uptake to preserve the sexual and reproductive rights of clients. Future studies on reasons for contraceptives choices and access to second trimester induced abortion is of essence.

## Supporting information

**S1 Fig. Graph showing the annual proportion of the various contraceptive uptake within the facility.**
(TIF)

**S1 Table. Annual trends in number of abortions stratified by age groups and gestational age.**
(DOCX)

**S2 Table. Trends in the contraceptive mix among women who had induced abortion (2010–2020).**
(DOCX)

**S3 Table. Actual and predicted values of abortion using linear regression modelling.**
(DOCX)

## Acknowledgments

We wish to express our profound gratitude to the chief executive officer of CMJAH, the Head of department of Obstetrics and Gynecology (CMJAH), all the staff of the department of obstetrics and gynecology especially those in ward 195.

## Author Contributions

**Conceptualization:** Kennedy Baffour-Duah, Lusanda Shimange-Matsose.

**Data curation:** Kennedy Baffour-Duah.

**Formal analysis:** Kennedy Baffour-Duah, Gbenga Olorunfemi.

**Funding acquisition:** Kennedy Baffour-Duah.

**Investigation:** Kennedy Baffour-Duah, Gbenga Olorunfemi, Lusanda Shimange-Matsose.

**Methodology:** Kennedy Baffour-Duah, Gbenga Olorunfemi, Lusanda Shimange-Matsose.

**Project administration:** Kennedy Baffour-Duah.

**Resources:** Kennedy Baffour-Duah.

**Software:** Gbenga Olorunfemi.

**Supervision:** Gbenga Olorunfemi, Lusanda Shimange-Matsose.

**Validation:** Kennedy Baffour-Duah, Gbenga Olorunfemi, Lusanda Shimange-Matsose.

**Visualization:** Kennedy Baffour-Duah, Gbenga Olorunfemi.

**Writing – original draft:** Kennedy Baffour-Duah.

**Writing – review & editing:** Gbenga Olorunfemi, Lusanda Shimange-Matsose.

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
