## [Decision Letter · Decision Letter 0]

22 Aug 2022

PONE-D-22-14781Pre- and intra -covid trends and pattern of contraceptive use among women who had termination of pregnancy at Charlotte Maxeke Johannesburg Academic Hospital, Johannesburg South Africa (2010-2020).PLOS ONE

Dear Dr. Dr. Kennedy Baffour, MD,

Thank you for submitting your manuscript to PLOS ONE. After careful consideration, we feel that it has merit but does not fully meet PLOS ONE’s publication criteria as it currently stands. Therefore, we invite you to submit a revised version of the manuscript that addresses the points raised during the review process.

ACADEMIC EDITOR: 

 The topic of the manuscript is interesting. Nevertheless, the reviewers raised several concerns: considering this point, I invite authors to perform the required major revisions.

A# Abstract

A. Methods

1. line 25, there is punctuation error and modified like this, annual contraceptive uptake rate (stratified by types)…..

2. line 28 also you missed punctuation, correct it.

B. Results

1. In line 30, the number is not clear to the reader. Did you say that 49.26%?

2.line3 5-36, you make …… annum from 53.45 per a hundred clients in 2014 to 98 per 100 hundred clients. You can change from……53.45 per 100 clients in 2014 to 98 per 100……

what is your cut point to say the majority ……you should rewrite again.

C. Introduction

1. Your introduction is too much long, shorten it.

2. Line 50 & 60, there is a grammar error, Contraception is a very important primary prevention strategy for maternal mortality……

3. line 54 too late information, we are in 2022. so, you included the updated information.

4. Line 57-58, you should correct it like this. It is expected that liberalization of access to induced abortion services can reduce unsafe abortion and its complications.

5. line 61, spelling error, line 62, punctuation error, line 65-68, need a reference, line 78, you said that prevalence rate, you should use either one. Prevalence is quite different to rate, and line 1oo punctuations.

D. Methods

1. Line 114, is the province city a capital city? revised it. Line 121-122 grammatically isn’t correct.

2. line 145, do have waivers for identified cases?

E. Result

1. Line, 208 grammatically isn’t correct, 219-226, needs modification, first you should write text, then followed by a table and figure. You should address all tables and figures based on the above comments.

F. discussion

1. Line,460, is number (226) or percent?

Please submit your revised manuscript by  October . If you will need more time than this to complete your revisions, please reply to this message or contact the journal office at plosone@plos.org. Please include the following items when submitting your revised manuscript:

We look forward to receiving your revised manuscript.

Kind regards,

Zemenu Yohannes Kassa, Msc

Academic Editor

PLOS ONE

Journal Requirements:

Reviewers' comments:

Reviewer's Responses to Questions

**Comments to the Author**

1. Is the manuscript technically sound, and do the data support the conclusions?

Reviewer #1: Yes

Reviewer #2: Yes

2. Has the statistical analysis been performed appropriately and rigorously? 

Reviewer #1: Yes

Reviewer #2: Yes

3. Have the authors made all data underlying the findings in their manuscript fully available?

Reviewer #1: Yes

Reviewer #2: Yes

4. Is the manuscript presented in an intelligible fashion and written in standard English?

Reviewer #1: Yes

Reviewer #2: Yes

5. Review Comments to the Author

Reviewer #1: Title

The short title seems better. It is more concise.

Abstract

Background

Lines 16-17: ‘The pattern of contraceptive uptake can guide interventions among these group of vulnerable individuals.’ The authors use ‘these group…’ without prior mention of any specific group

Methods:

Use uppercase for COVID-19. Write it in full at first use.

Results

The is no report on associations in the results

Introduction

Lines 70-72: ‘Furthermore, the 1996 constitution of the republic of South African, Act 108 of 1996, promotes reproductive rights and the rights of access to reproductive health care’. Is this ACT available online? If yes, could you include a reference for readers interested in obtaining further information about this ACT?

Lines 74, 76, 90-91: Some grammatical errors noted; government supplies’, ‘the contraceptive prevalence rate… remains’, use of ‘although’ and ‘however’ in same sentence.

Methods:

The authors should consider moving the ‘Ethical consideration’ section to the end of methods section.

Line 161: typo noted; joint

Statistical analysis

Two methods were used to predict the expected numbers for 2020. What is the rationale for using both? What advantages do the methods offer, over other prediction models?

Did the trend analysis account for changes in underlying at-risk population? It is highly likely that the population in catchment area may have varied from year to year.

Results

Line 270: Grammatical error noted; the uptake started increasing remained high until 2015….

Table 3 column 4 row 3: Error in reported figure

There is no figure 4.

Discussion

Line 419: Grammatical error noted; contraceptive uptake has increased to about 98% …

Line 514: Grammatical error noted; annual number of induced abortions

Reviewer #2: Overall, the manuscript addresses a very important subject in the area of reproductive health.

The research questions and key findings are well presented by the authors.

Overall, the claims by the authors are grounded in the data presented.

I have a minor comments for the authors to address:

1. The authors should clarify whether the study was able to determine the types of contraceptives available at the study site. If no data is available to support this claim, then they cannot recommend that "other long-acting contraceptives should be made available" when they have not ascertain the currently available options.

2. The authors should do a thorough a proof reading of the manuscript and ensure consistency in the use of terminologies. eg. Covid-19 etc.

6. PLOS authors have the option to publish the peer review history of their article (what does this mean?). If published, this will include your full peer review and any attached files.

Reviewer #1: No

Reviewer #2: No

---

## [Author Response · Author response to Decision Letter 0]

6 Oct 2022

We are very grateful to the Editor and the two reviewers for their comments on our manuscript. Below is our response to the comments raised by the academic registrar and the two reviewers. We have tried to address all the comments raised and believe that our paper has really improved considerably. We would also be happy to make further corrections to our paper if necessary.

We hope our response would be granted the needed attention and consideration it deserves.

Yours Sincerely,

On behalf of all the authors

Dr Baffour-Duah.

ACADEMIC EDITOR:

A # Abstract

A.Methods

1.Line 25, there is a punctuation error and modified like this, annual contraceptive uptake rate(stratified by types)

 The sentence was modified and the punctuation error addressed. We are very grateful.

2.Line 28 also you missed punctuation, correct it.

 We have corrected the punctuation omission. Thank you very much

B.Results 

1.In line 30, the number is not clear to the reader. Did you say that 49.26%?

We have clarified the statement. Yes we meant 49.26%. Thank you

2.Lines 35-36, you make…. annum from 53.45 per a hundred clients in 2014 to 98 per 100 hundred clients. You can change from ….53.45 per 100 clients in 2014 to 98 per 100….

We have effected the changes. Thank you.

What is your cut point to say the majority……. you should rewrite again

We have addressed the statement and replaced the majority with most.

C.Introduction

1.Your introduction is too long, shorten it

We have worked on the introduction. We have tried to focus on the key issues related to the content of the manuscript. We are very grateful.

2.Line 50 & 60, there is a grammar error, contraception is a very important prevention strategy for maternal mortality…..

We have edited that statement. Thank you

3.Line 54 too late information, we are in 2022, so, you included the updated information

We have removed the entire statement to also help in reducing the content of the introduction. Thank you.

4.Line 57-58, you should correct it like this. It is expected that the liberalization of access to induced abortion services can reduce unsafe abortion and its complications.

We have addressed the comment and effected the change.

5.In line 61, spelling error, line 62, punctuation error, line 65-68, need a reference, line, 78, you said that the prevalence rate, you should use either one. Prevalence is quite different to rate, and in line 100 punctuation

We have addressed all the punctuation errors, referenced lines 65-68 and remove the rate from the sentence.

D.Methods

1.Line 114, is the province city a capital city? Revise it. Line 121-122 grammatically isn’t correct. 

Johannesburg is not the capital city of South Africa. It is a provincial capital of Gauteng and the largest city in south Africa. We have addressed this and the grammatical errors as well. Thank you.

2.Line 145, do have waivers for identified cases?

We did get approval from the hospital CEO and from WITS university HREC. We only analysed the records available . confidentiality of the data was ensured as we didn’t use names or other identifiers. The study was a retrospective cross-sectional study. We reviewed the nursing records and summaries of the induced abortion cases that were performed for the past eleven years from 1 January 2010 to 31 December 2020. The annual number of patients who presented for termination of pregnancy, annual number of patients who had contraception afterwards and the various methods that were accepted were extracted into a spreadsheet for analysis. Confidentiality of the data was ensured as names and other identifier were not utilized. The data obtained was aggregated and fully anonymized. The ethics committee waved the requirements for informed consent from the clients to have the data from the nursing records used in the research.

E.Results

1.Line 208 grammatically inst correct. 219-226, needs modification, first you should write test, then followed by a table and figure. You should address all tables and figures based on the above comments.

we corrected the grammatical error, amended the text and table arrangements to comply with the style requirements.

F.Discussion 

1.Line 460, is number (226) or percent

The 226 is a number not a percent. Thank you.

REVIEWER #1

Title 

The short title seems better, it is more concise

We are very grateful.

Abstract

Background

Lines 16-17:” the pattern of contraceptive uptake can guide interventions among these group of vulnerable individual”. the authors use “these group” without prior mention of any specific group

We have amended the statement and removed these group.

Methods

Use uppercase for COVID-19. write it in full at first use.

We have addressed this in the manuscript. Thank you.

Results

There is no report on association in the results.

We noted this and have corrected it in manuscript. Thank you.

Introduction

Lines 70-72:” Furthermore, the 1996 constitution of the republic of South Africa, Act 108 of 1996, promotes reproductive rights and the rights of access to reproductive health care.” is this ACT available online? If yes, could you include a reference for readers interested in obtaining further information about this ACT?

Most of the ACTS in the South African 1996 constitution are available online including ACT 108. We have included a reference in this context with regards to the promotion of reproductive rights and the rights of access of reproductive health care. Thank you for drawing our attention to this.

Lines 74,76,90-91: some grammatical errors noted: government supplies; the contraceptive prevalence rate …remains; use of although and however in same sentence.

We thank you for drawing our attention to them. We have made the necessary corrections.

Methods:

The authors should consider moving the ethical considerations section to the end of the methods section

The ethical consideration section follows the study procedure which ends the method section. Thank you.

Line 161: typo noted: joint

We have corrected it to Join. Thank you

Statistical analysis

Two methods were used to predict the expected numbers for 2020. what is the rationale for using both? What advantages do the methods offer, over other prediction models?

Indeed there are many prediction modelling that have been described in the literature. However, join point regression was utilised as a novel tool for predicting trends. We then utilised the linear regression models to compare results and we still had similar outcomes. Thank you

Did the trend analysis account for changes in underlying at-risk population? It is highly likely that the population in catchment area may varied from year to year.

we thank the reviewer for this comment. Our analysis was on trends in contraceptive uptake rates among abortion cases. Since we utilised rates per abortion cases and not the actual numbers of contraceptive uptake, we believe that the trends analysis had corrected for the at risk population who were the clients that came for abortion services. Thus changes in the abortion clients who were the at risk population will also reflect in our trends.

Results :

Line 270: grammatical error noted; the uptake started increasing remained high until 2015….. table 3 column 4 row 3: error is reported in figure. There is no figure 4

Grammatical errors noted, corrections made. Figure 4 has been corrected and the error in table 3 rectified now. Thank you. 

Discussion:

Line 419: grammatical error noted; contraceptive uptake has increased to about 98%…… Line 514: grammatical error noted; annual number of induced abortions

The corrections have been made. Thank You

Reviewer #2

Overall , the manuscript addresses a very important subject in the area of reproductive health

We are very grateful.

The research questions and the findings are well presented by the authors

We are very grateful.

---

## [Decision Letter · Decision Letter 1]

31 Oct 2022

PONE-D-22-14781R1Pre- and intra -COVID-19 trends and pattern of contraceptive use among women who had termination of pregnancy at Charlotte Maxeke Johannesburg Academic Hospital, Johannesburg South Africa (2010-2020).PLOS ONE

Dear Dr. Kennedy Baffour-Duah, MD,

Thank you for submitting your manuscript to PLOS ONE. After careful consideration, we feel that it has merit but does not fully meet PLOS ONE’s publication criteria as it currently stands. Therefore, we invite you to submit a revised version of the manuscript that addresses the points raised during the review process.

We look forward to receiving your revised manuscript.

Kind regards,

Zemenu Yohannes Kassa, Msc

Academic Editor

PLOS ONE

Journal Requirements:

Additional Editor Comments (if provided):

Reviewers' comments:

Reviewer's Responses to Questions

**Comments to the Author**

1. If the authors have adequately addressed your comments raised in a previous round of review and you feel that this manuscript is now acceptable for publication, you may indicate that here to bypass the “Comments to the Author” section, enter your conflict of interest statement in the “Confidential to Editor” section, and submit your "Accept" recommendation.

Reviewer #1: All comments have been addressed

2. Is the manuscript technically sound, and do the data support the conclusions?

Reviewer #1: Yes

3. Has the statistical analysis been performed appropriately and rigorously? 

Reviewer #1: Yes

4. Have the authors made all data underlying the findings in their manuscript fully available?

Reviewer #1: Yes

5. Is the manuscript presented in an intelligible fashion and written in standard English?

Reviewer #1: Yes

6. Review Comments to the Author

Reviewer #1: The authors have addressed the initial comments while maintaining the original meaning of the manuscript. The manuscript is of suitable standard for publication. I do have one minor issue.

Lines 94-97

'ALTHOUGH, virtually all contraceptive technologies can be safely utilized by clients as post abortion

contraceptives, BUT the pattern of contraceptive uptake may be influenced by health system

concerns such as distribution and availability of the various contraceptive types, and the individual

experiences and preference'

I suggest that you delete 'ALTHOUGH' or 'BUT'

7. PLOS authors have the option to publish the peer review history of their article (what does this mean?). If published, this will include your full peer review and any attached files.

Reviewer #1: No

---

## [Author Response · Author response to Decision Letter 1]

31 Oct 2022

Department of Obstetrics and Gynaecology

University of Witwatersrand

Johannesburg, South Africa

31st October, 2022.

The Editor in Chief,

PLOS ONE Journal

REBUTTAL LETTER

We are very grateful to the Editor and the reviewers for their comments on our manuscript. Below is our response to the comment raised by the academic registrar and the reviewers. We have tried to address the comments raised and believe that our paper has really improved considerably.

We hope our response would be granted the needed attention and consideration it deserves.

Yours Sincerely,

On behalf of all the authors

Dr Baffour-Duah.

Reviewer #1

Lines 94-97

‘ ALTHOUGH, virtually all contraceptive technologies can be safely utilized by clients as post abortion contraceptives, BUT the pattern of contraceptive uptake may be influenced by health system system concerns such as distribution and availability of the various contraceptive types, and the individual experiences and preference’

I suggest that you delete ‘ALTHOUGH’ or ‘BUT’

We have changed the sentence. Thank you.

---

## [Editor Report · Decision Letter 2]

6 Nov 2022

Pre- and intra -COVID-19 trends and pattern of contraceptive use among women who had termination of pregnancy at Charlotte Maxeke Johannesburg Academic Hospital, Johannesburg South Africa (2010-2020).

PONE-D-22-14781R2

Dear Dr. Baffour-Duah,

We’re pleased to inform you that your manuscript has been judged scientifically suitable for publication and will be formally accepted for publication once it meets all outstanding technical requirements.

Kind regards,

Zemenu Yohannes Kassa, Msc

Academic Editor

PLOS ONE

Additional Editor Comments (optional):

In the title , you can remove "pattern" or you should give clear operational definition

#Abstract

Remove pattern line 16,18, introduction 95,98,102,388,533

#Result

you used "most " what is your cut off point to say most or majority , please rewrite again.line 505 ,71% .....some times you said 66% ,think over it in this word.

#conclusion

you should remove "Majority of the clients accepted injectable contraceptives. " this sentence. 538 and 539.

#introduction

line 64 ....important primary....either use one important or primary .it is similar meaning. remove one .line 69,and 94 modify contraceptive methods, line 78 contraceptive methods, 307,331

you should put a operational definition for unintended and unwanted pregnancy or you can use one word across your manuscript. line 413

#methods

line 122 midwives,176 it needs citation,

#Result

209 remove pattern
---

## [Editor Report · Acceptance letter]

18 Nov 2022

PONE-D-22-14781R2 

Pre- and intra -COVID-19 trends of contraceptive use among women who had termination of pregnancy at Charlotte Maxeke Johannesburg Academic Hospital, Johannesburg South Africa (2010-2020). 

Dear Dr. Baffour-Duah:

I'm pleased to inform you that your manuscript has been deemed suitable for publication in PLOS ONE. Congratulations! Your manuscript is now with our production department. 

Kind regards, 

on behalf of

Dr. Zemenu Yohannes Kassa 

Academic Editor

PLOS ONE